🔓 | **Open Peer Review** | Computational Biology | Research Article

# Functional metabolomics of the human scalp: a metabolic niche for *Staphylococcus epidermidis*

Louis-Félix Nothias,[1,2] Robin Schmid,[1,2,3] Allison Garlet,[4] Hunter Cameron,[5] Sabrina Leoty-Okombi,[6] Valérie André-Frei,[6] Regine Fuchs,[7] Pieter C. Dorrestein,[1,2] Philipp Ternes[7]

**ABSTRACT** Although metabolomics data acquisition and analysis technologies have become increasingly sophisticated over the past 5–10 years, deciphering a metabolite's function from a description of its structure and its abundance in a given experimental setting is still a major scientific and intellectual challenge. To point out ways to address this "data to knowledge" challenge, we developed a functional metabolomics strategy that combines state-of-the-art data analysis tools and applied it to a human scalp metabolomics data set: skin swabs from healthy volunteers with normal or oily scalp (Sebumeter score 60–120, $n = 33$; Sebumeter score > 120, $n = 41$) were analyzed by liquid chromatography-tandem mass spectrometry (LC-MS/MS), yielding four metabolomics data sets for reversed phase chromatography (C18) or hydrophilic interaction chromatography (HILIC) separation in electrospray ionization (ESI) + or − ionization mode. Following our data analysis strategy, we were able to obtain increasingly comprehensive structural and functional annotations, by applying the Global Natural Product Social Networking (M. Wang, J. J. Carver, V. V. Phelan, L. M. Sanchez, et al., Nat Biotechnol 34:828–837, 2016, https://doi.org/10.1038/nbt.3597), SIRIUS (K. Dührkop, M. Fleischauer, M. Ludwig, A. A. Aksenov, et al., Nat Methods 16:299–302, 2019, https://doi.org/10.1038/s41592-019-0344-8), and MicrobeMASST (S. ZuffaS, R. Schmid, A. Bauermeister, P. W, P. Gomes, et al., bioRxiv:rs.3.rs-3189768, 2023, https://doi.org/10.21203/rs.3.rs-3189768/v1) tools. We finally combined the metabolomics data with a corresponding metagenomic sequencing data set using MMvec (J. T. Morton, A. A. Aksenov, L. F. Nothias, J. R. Foulds, et. al., Nat Methods 16:1306–1314, 2019, https://doi.org/10.1038/s41592-019-0616-3), gaining insights into the metabolic niche of one of the most prominent microbes on the human skin, *Staphylococcus epidermidis*.

**IMPORTANCE** Systems biology research on host-associated microbiota focuses on two fundamental questions: which microbes are present and how do they interact with each other, their host, and the broader host environment? Metagenomics provides us with a direct answer to the first part of the question: it unveils the microbial inhabitants, e.g., on our skin, and can provide insight into their functional potential. Yet, it falls short in revealing their active role. Metabolomics shows us the chemical composition of the environment in which microbes thrive and the transformation products they produce. In particular, untargeted metabolomics has the potential to observe a diverse set of metabolites and is thus an ideal complement to metagenomics. However, this potential often remains underexplored due to the low annotation rates in MS-based metabolomics and the necessity for multiple experimental chromatographic and mass spectrometric conditions. Beyond detection, prospecting metabolites' functional role in the host/microbiome metabolome requires identifying the biological processes and entities involved in their production and biotransformations. In the present study of the human scalp, we developed a strategy to achieve comprehensive structural and functional annotation of the metabolites in the human scalp environment, thus diving

Address correspondence to Philipp Ternes, philipp.ternes@basf.com.

P.C.D. is an advisor to Cybele and co-founder and scientific advisor to Ometa and Enveda, with prior approval by UC San Diego.

See the funding table on p. 17.

one step deeper into the interpretation of "omics" data. Leveraging a collection of openly accessible software tools and integrating microbiome data as a source of functional metabolite annotations, we finally identified the specific metabolic niche of *Staphylococcus epidermidis*, one of the key players of the human skin microbiome.

**KEYWORDS**   metabolomics, skin microbiome, scalp, metabolite annotation, multi-omics integration

The advent of "omics" technologies underscored the intricate metabolic relationship between living organisms and their associated microorganisms, collectively termed the microbiota. One of the pivotal "omics" technologies, metabolomics, offers a deep dive into the metabolic profiles of these organisms and their environment, revealing the dynamic interplay of metabolites that shape their interactions. Mass spectrometry (MS)-based metabolomics, in particular, has evolved to provide a panoramic view of these metabolites, reflecting both microbial and host metabolic activities. This interplay, termed meta-metabolism, encompasses the cellular pathways that dictate function and the metabolic processes that mediate interspecies interactions.

Their symbiotic association with the host, involving either mutualism or parasitism, is called a holobiont. Understanding the molecular processes and functions implicated in holobionts could be translated into biomimetic clinical applications (1). For instance, the skin-associated microbiota has emerged as a pivotal player in the onset and prevention of various skin conditions (2). The microbial diversity of the human skin, exemplified by the protective role of *Staphylococcus epidermidis* against skin neoplasia (3), is a testament to its potential in maintaining skin equilibrium.

Advancements in microbiome sequencing techniques have enabled the direct profiling of microbial species and their genomes directly from diverse samples, transcending the constraints of traditional lab-based cultivation techniques. Shotgun metagenomic sequencing further refined this approach, shedding light on predominant metabolic pathways, even with specific microorganisms. However, the actualized genetic potential of microorganisms within a holobiont is modulated by a myriad of factors, including microbial dynamics, host genetics, and environmental influences. These intricate molecular dialogs can be captured through transcriptomics, proteomics, and, notably, metabolomics analyses.

Furthermore, metabolomics complements other omics technologies by highlighting small molecules potentially resulting from complex metabolic processes, either biotic or abiotic, often elusive to other omics technologies. In addition, modern MS-based metabolomics can now survey a vast array of metabolites by collecting their fragmentation spectra (MS/MS) in a high-resolution and data-dependent manner, which yields fragmentation fingerprints for metabolite annotation (4).

Yet, challenges persist. The annotation of metabolites from MS/MS spectra has historically been hampered by the lack of reference spectra in spectral libraries (typically offering less than 5%–10% of the annotation rates) (5). The efficacy of mass spectrometry in detecting metabolites is contingent upon the specific chromatographic and ionization methods employed. Furthermore, synthesizing data from diverse omics platforms remains a formidable challenge. Although machine learning models have been introduced to hypothesize microbe-metabolite interactions (6), the precise functional role of the metabolite is challenging to interpret.

In this study, we charted a comprehensive strategy to explore functions of the meta-metabolism in holobionts. Metabolomics analysis was performed in multiple complementary mass spectrometry experimental conditions to ensure comprehensive coverage of diverse metabolites, and their structural annotation was made possible thanks to the latest advances in computational mass spectrometry methods (7), including Global Natural Product Social Networking (GNPS) spectral library searches (8), and systematic chemical class annotation with SIRIUS/CANOPUS (9). Functional metabolite annotation was initially performed by correlating

liquid chromatography-tandem mass spectrometry (LC-MS/MS) features with physical parameters of the scalp (oiliness and moisture), as well as by searching for matching spectra in publicly available data sets from microbial monocultures using microbe-MASST (10). Functional metabolite annotation was further enhanced by integrating the metabolomics data with microbial taxon abundances derived from metagenomic sequencing, predicting conditional probabilities of microbe-metabolite associations using a neural network framework (6). Structural and functional metabolite annotations were displayed as richly annotated molecular networks with Cytoscape (11) and were used to generate hypotheses on their function in the microbial and host-microbe interactions.

## RESULTS AND DISCUSSION

### A functional metabolomics strategy

The functional metabolomics strategy comprises four steps (Fig. 1), which were addressed by the following approaches: (i) mass spectrometry-based metabolomics (LC-MS/MS with two chromatographic conditions and two ionization modes), (ii) structural metabolite annotation (library-based annotations and molecular networking with GNPS; *in silico* annotations with SIRIUS), (iii) functional metabolite annotation (statistical analysis of physical skin parameters, search for microbial metabolites with microbeMASST, and discovery of microbe–metabolite co-occurrence patterns with MMvec, and (iv) visualization and interpretation of results (rich visualization of molecular networks with Cytoscape; word-cloud visualization of compound classes). This functional metabolomics strategy can be applied with biological or environmental samples in various experimental settings. This is made possible by a large and still growing ecosystem of publicly available metabolite annotation tools. Notably, these tools allow researchers to empirically annotate features in untargeted metabolomics data sets without prior molecular knowledge or hypothesis. Compelling is the possibility to derive functional metabolite annotations by bringing in data from other "omics" technologies, in the present study, metagenomic sequencing.

### Mass spectrometry-based metabolomics

Multiple untargeted mass spectrometry analyses were performed to cover a wide chemical space of metabolites differing between normal and oily scalps. First, the metabolites were sampled on 74 healthy volunteers (normal scalp, *n* = 33; oily scalp,

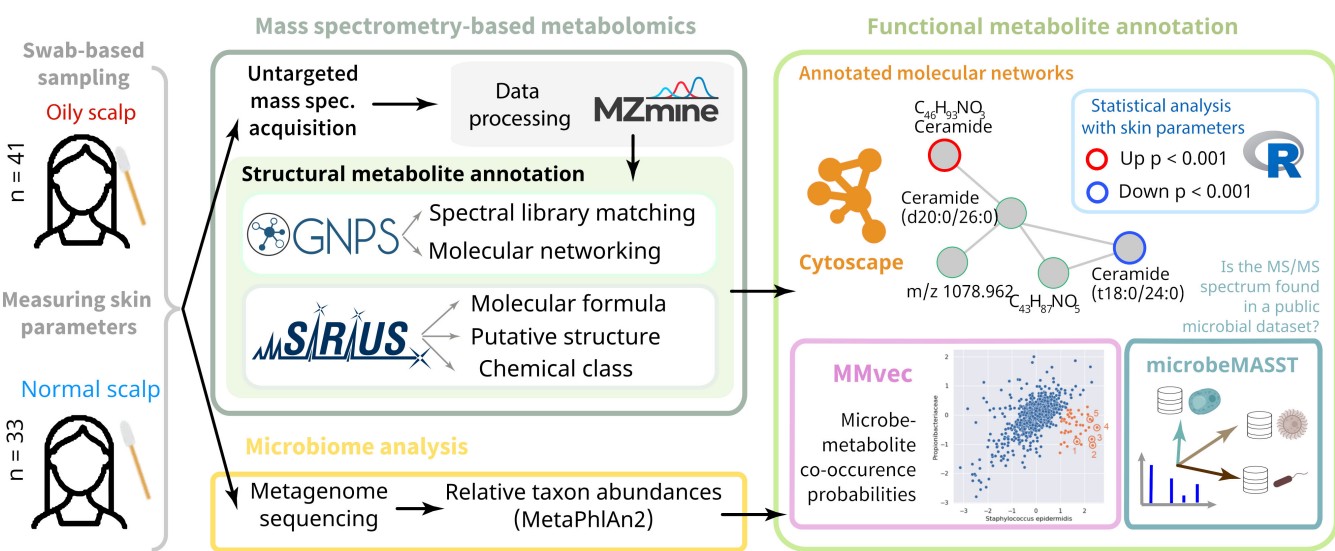

**FIG 1** Overview of the study design and multi-omics analysis strategy relying on both untargeted metabolomics and metagenomics sequencing. The "omics" integration was performed with MMvec, and the results were visualized as a microbe-metabolite network.

*n* = 41; see Materials and Methods for details) using ethanol-pre-soaked cotton swabs, as previously described (12). To cover a broad range of metabolites, each sample was analyzed by LC-MS/MS under four experimental conditions: two chromatographic methods (reversed-phase chromatography [C18] and hydrophilic interaction chromatography [HILIC]) and two ionization modes (positive and negative mode electrospray ionization [ESI]). The data are available as a MassIVE data set (MSV000088283, doi:10.25345/C5DK1G)

The data from each of these conditions were processed separately with MZmine (13), using the Ion Identity Molecular Networking (IIMN) workflow for improved identification of ion types (e.g., adducts and in source fragments) by feature shape correlation analysis (14). Briefly, ion features (*m/z*, retention time, isotope pattern, and MS/MS) are detected and aligned across all samples, grouped with their corresponding MS/MS spectra, and exported for SIRIUS and the IIMN workflow on the GNPS web platform (8). The fragmentation spectra were then annotated by spectral library search against the reference spectra from the GNPS open-community libraries before molecular networking by a pair-wise modification-aware spectral similarity of all MS/MS (15). Optionally, analog compounds were annotated by modification-aware library search.

To expand the annotation coverage, the spectral data from all the metabolites detected were further annotated with SIRIUS (4), providing putative annotations for (i) molecular formulas (16), (ii) putative structures (17), and (iii) chemical class (9). The results of GNPS and SIRIUS were combined and visualized as annotated molecular networks. The annotation of the molecular networks was further enriched with statistical analysis results by displaying the correlation of feature intensity (chromatographic peak area, log scale) with the skin's oiliness (Sebumeter value).

## Metabolite annotations from spectral libraries and molecular networking with GNPS

To examine the annotation coverage obtained by spectral library matching with the GNPS and the NIST spectral libraries, the annotated features and their distribution in data sets were visualized as an UpSet plot (Fig. 2A). HILIC provided the largest number of library hits (286 matches) in the positive ionization mode (HILIC+) but only 84 matches in the negative ionization mode (HILIC−). C18 gave 105 matches in the positive ionization mode (C18+) and 21 matches in the negative ionization mode (C18−). The highest overlap of spectral library hits was observed between the HILIC+ and C18+ (78 matches) and between the HILIC+ and HILIC− (57 matches). Overall, these results illustrate that each metabolomics data set contributes to a different metabolic space, depending on the metabolites' polarity and ionization preference.

When comparing the number of annotations to the total number of features, the annotation rate was 2%–4% in each data set (Fig. 2B; Fig. S1; Table S1). This proportion was increased to 9%–10% by performing spectral library matching in the analog mode, which tolerates a spectral shift for the fragment and precursor ions between the experimental and the library spectra. Yet, this analog mode is known to have a higher chance of providing incorrect matches. For molecular networking results, the number of nodes in networks was lower for the HILIC+ (16%) and HILIC− (27%) data sets and higher for the C18+ (29%) and C18− (53%) data sets. These differences are likely due to different fragmentation behaviors for chemical classes preferentially observed in each data set.

## Expanding metabolite annotations with SIRIUS

To increase the annotation coverage, we computationally annotated molecular formulas (16), putative structures (17), and chemical classes (9) using the SIRIUS tools. As observed for the spectral library annotation and molecular networking results, the proportion of annotations with SIRIUS varied per data set (Fig. 2B; Table S1). Nevertheless, the annotation rate was considerably expanded for all data sets, with a 23.5-fold increase in annotation coverage for the C18+, a 45-fold increase for C18−, a 7.6-fold increase for HILIC+ and a 13.5-fold increase for HILIC−. Chemical class predictions were obtained with

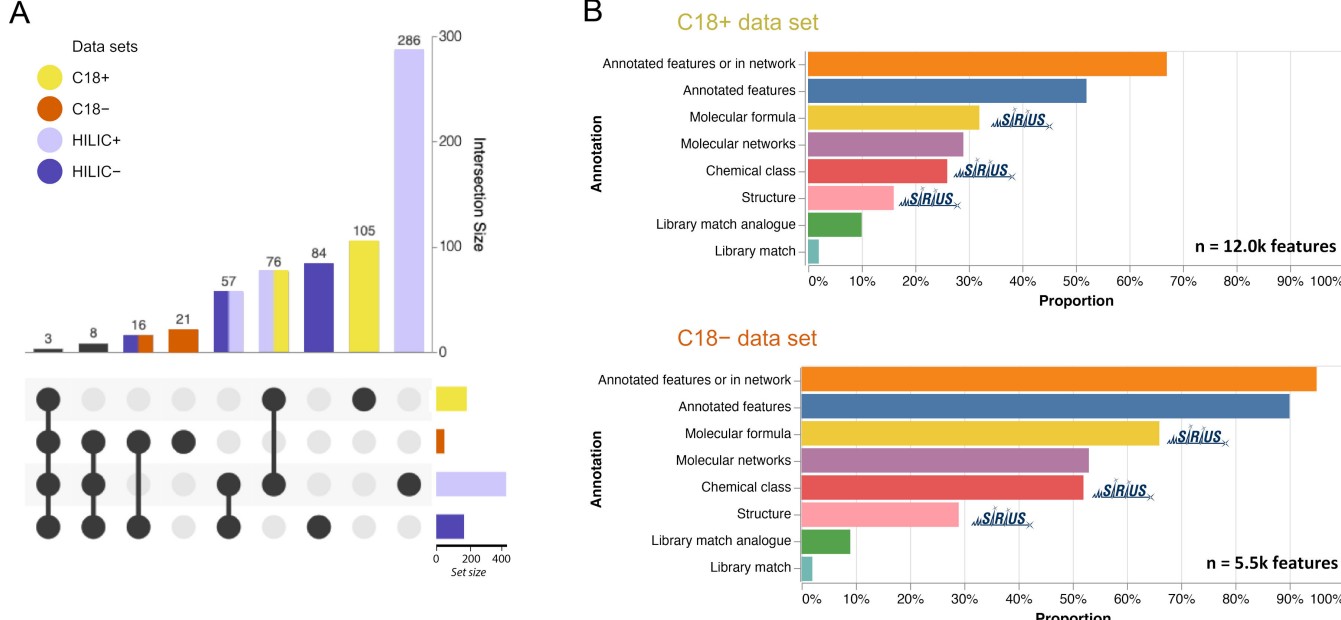

FIG 2  (A) Distribution of metabolites annotations per data set (C18+, C18−, HILIC+, and HILIC− data sets) and (B) proportion of LC-MS features annotated by GNPS or SIRIUS in the C18+ and C18− data sets (top, positive ionization mode; bottom, negative ionization mode).

CANOPUS for 26% and 29% of the features in the C18+ and C18− data sets, respectively. The rate was lower in the HILIC data sets, with 16% and 5% for the HILIC+ and HILIC−, respectively. The source of this difference is likely due to the machine-learning nature of the class prediction method. Indeed, the number of reference spectra available in public spectral libraries acquired in the negative ionization mode or observed in HILIC is relatively limited. Thus, fewer reference spectra were available to train the CANOPUS neural network, resulting in fewer class predictions for HILIC−. For putative structure prediction with CSI:FingerID, the annotation rate ranged from 11% (HILIC+) to 30% (HILIC−). Overall, compared with spectral library matching-based annotation, CSI:FingerID increased the network annotation rate by 1.75 (C18+ and HILIC−) to 15.5-fold (C18−).

When combining the annotations from spectral library matching (both direct and analog modes) with the SIRIUS annotations, the total feature annotation rate reached 52%, 90%, 24%, and 54% for the C18+, C18−, HILIC+, and HILIC− data sets, respectively. Furthermore, when combining these annotations with molecular networking, 67% (+22.4%), 95% (+5.3%), 28% (+14.3%), and 57% (+5.3%) of the features were either directly annotated or connected in an annotated molecular network. This illustrates how combining multiple annotation strategies allows us to better apprehend the chemical diversity observed in metabolomics data sets.

## Evaluating the consistency of the SIRIUS predictions

We relied on the following methodology to evaluate the consistency of the data processing and annotation performed on each data set: high-quality spectral library matches were considered ground truth for comparing SIRIUS-based molecular formula and class prediction.

Results showed that the predicted molecular formula matched the spectral library annotation in 66% (44 matches), 70% (10 matches), 90% (125 matches), and 95% (59 matches) of evaluation pairs for the C18+, C18−, HILIC+, and HILIC− data sets, respectively (Fig. S2 through S5). Inspection of incorrect molecular formula predictions revealed that an incorrect adduct assignment was often involved (in particular, sodiated instead of protonated for the positive ionization mode). This is also supported by the higher

annotation consistency for the negative ionization. In addition, the lower consistency rate in C18 data sets could also be related to the high number of isobaric lipid species observed in this chromatographic condition, which are challenging to annotate unambiguously, even in high-resolution mass spectrometry.

An assessment of the chemical class prediction accuracy was not feasible for the data sets in the negative ionization mode, as only 10 pairs of high-confidence spectral library and chemical class prediction were available for each. However, 44 and 120 pairs were available in the positive ionization mode for C18+ and HILIC+, respectively. The chemical class was correctly predicted for C18+ and HILIC+ data sets, respectively, at the following accuracy rates: 84% and 92%, respectively, for the superclass level, 80% and 90%, respectively, at the class level, and 57% and 86%, respectively, for the subclass level. Interestingly, this evaluation shows that the chemical class prediction was relatively robust for the C18+ data set, even if the molecular formula prediction accuracy was comparatively lower at 66%.

## Statistical analysis of physical skin parameters

We next assessed the correlation of each observed feature's intensity with physical skin parameters from the sample metadata. After a filtering step to remove features that contributed little to biological information (high feature intensities in blank samples or high technical variability), the feature intensities were correlated with scalp oiliness (assessed with a Sebumeter) and scalp moisture (skin conductance measured using a Skicon device) by analysis of variance (ANOVA). The statistical analysis comprised 44,570 features from the four data sets combined.

Figure 3 shows selected molecular networks from the C18+ (Fig. 3A) and HILIC+ (Fig. 3B) data sets. Symbols represent the annotation method of the feature, while colors highlight statistically significant differences between skin types. The molecular network in Fig. 3A contains several fatty acid methyl and ethyl esters as well as monoacylglycerols that are positively correlated with scalp oiliness (GNPS spectral library annotation, inverted triangle). Interestingly, nodes with a direct GNPS spectral library annotation consistently correlate positively with scalp oiliness, while most other nodes do not.

Fatty acid methyl esters and their sulfonated or ethoxylated derivatives are common constituents of cosmetic products (18). This may point to different use of cosmetic products by volunteers with oily scalps, e.g., more frequent hair washing. Although volunteers agreed to refrain from using shampoo or other cosmetics products starting 2 days before the first sample collection, some cosmetics ingredients were detected on the skin, showing that they perdure on the skin for several days (12).

Monoacylglycerols may originate from triacylglycerols, a major constituent of sebum, by the action of microbial lipases. Fatty acid ethyl esters are established products of microbial metabolism (19). Most of the available literature on bacteria and yeasts capable of producing fatty acid ethyl esters is focused on their biotechnological utilization for making biodiesel. We are unaware of any studies on their physiological role within the ecosystem of the human scalp. We speculate that in a lipid-rich environment like sebum, fatty acids (liberated from triacylglycerols by bacterial lipases) constitute a nearly unlimited carbon and energy source. In such an environment, the rate at which acetyl-CoA is being produced by microbial fatty acid β-oxidation may exceed its consumption in the TCA cycle. Reduction to ethanol followed by esterification to abundant fatty acids may constitute a safe way of disposing excess carbon while utilizing the NADH being produced during β-oxidation.

The network in Fig. 3B comprises nodes negatively correlated with scalp oiliness. While there are no GNPS annotations, the SIRIUS annotations are consistent with most nodes representing one or several classes of polar lipids containing one or three N in their molecular formula, e.g., lysosphingolipids. Sphingolipids are typical stratum corneum lipids. Thus, a decrease of sphingolipids or sphingolipid metabolites on oily scalp points to a shift in the relative proportions of sebum-derived and stratum corneum-derived metabolites. Overproduction of sebum by volunteers with oily scalps will

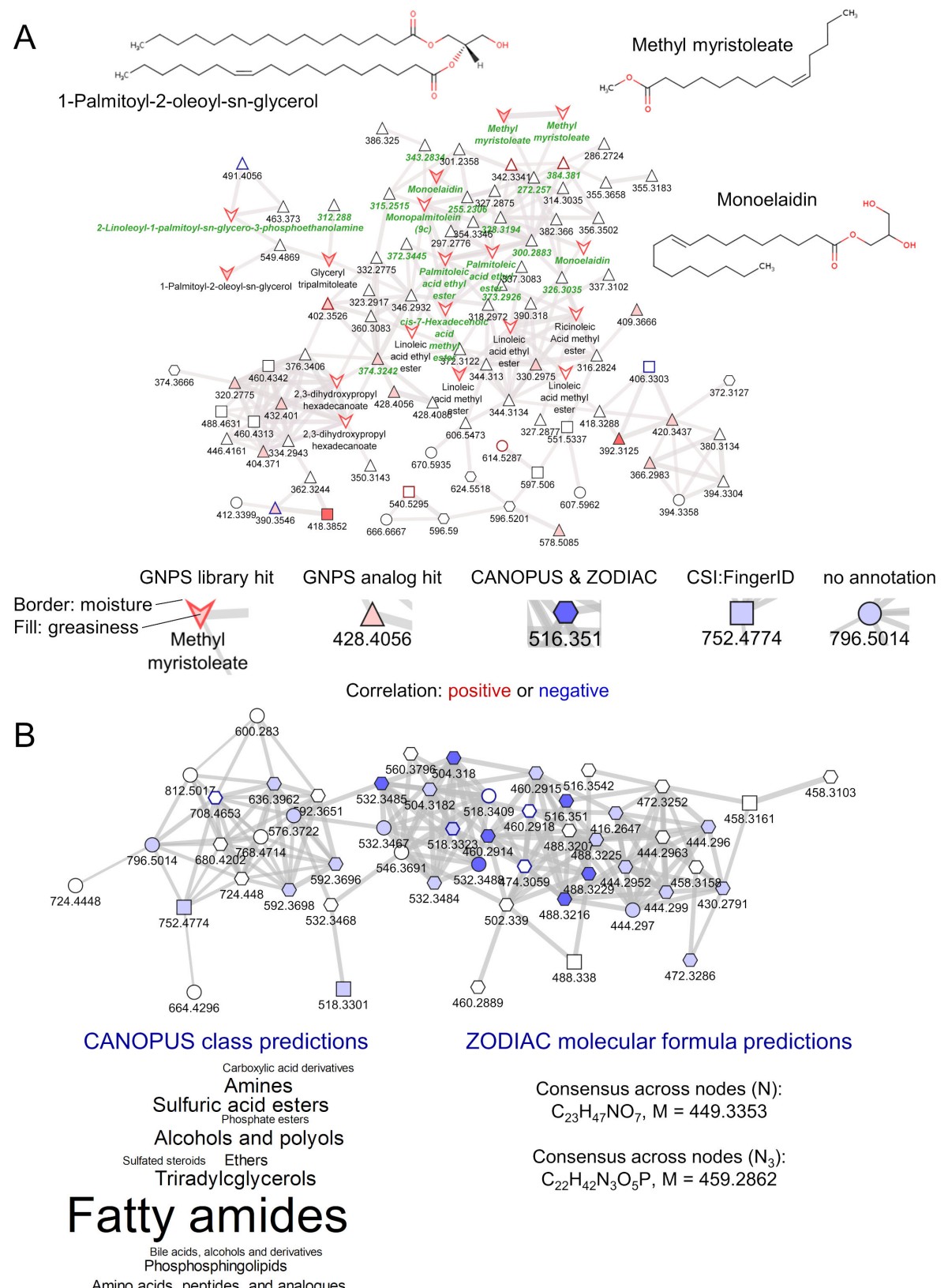

FIG 3   Molecular families. (A) Example from the C18+ data set showing structures for selected GNPS library spectral library matches. Green labels indicate a microbeMASST spectral match to a microbial monoculture. (B) Example from the HILIC+ data set showing consensus SIRIUS annotations. Node colors (red or blue) indicate the $P$-values for the correlation with oiliness (fill) or moisture (border): darker color, $P < 0.001$; medium color, $P < 0.01$; weak color, $P < 0.1$.

dilute stratum corneum-derived metabolites (20), thus changing the environment for the skin microbiota.

## Interim conclusions from the annotated metabolomics data

These examples show that biologically meaningful conclusions can be drawn from the metabolomics data already at the present analysis stage. But despite the richness in structural metabolite annotation achieved by the combination of library searches, molecular networking, and advanced *in silico* prediction tools, the results so far are descriptive in nature. Insight into the functioning of the skin ecosystem remained indirect or speculative. We suspected that this was because only a single functional annotation method was applied, namely, statistical analysis of physical skin parameters. We, therefore, asked whether additional, independent approaches for functional metabolite annotation would complement the statistical analysis and yield a higher level of biological insight. We further applied two complementary approaches, which both aim at empirically establishing links between metabolites and microbes: (i) systematic mass spectral search for features in the metabolomics data sets that were also detected in publicly available data sets from microbial monocultures using microbeMASST (10), a domain-specific MASST tool (21) that searches MS/MS data against data of ~60,000 taxonomically curated cultures of microbes (https://masst.ucsd.edu/microbemasst/) and (ii) discovery of microbe–metabolite co-occurrence patterns by combining the metabolomics data sets with a complementary metagenomic sequencing data set using MMvec (6).

## Search for microbial metabolites with microbeMASST

MASST (21) is a recent addition to the GNPS ecosystem and allows to search whether a fragmentation spectrum in a metabolomics data set has been previously observed in any data set from the public data repository part of the GNPS/MassIVE knowledge base. microbeMASST (https://masst.ucsd.edu/microbemasst/) performs a MASST search against a curated set of metabolomics data sets from microbial monocultures. To systematically link metabolites from the skin swabs to specific microorganisms, we searched all feature's fragmentation spectra from networks with at least three connected nodes with microbeMASST in batch mode.

The network shown in Fig. 3A was special in comprising many features with a positive correlation with scalp oiliness (red fill color) and many microbeMASST hits (green labels). Detailed annotations for each hit can be found in Fig. S6 through S26. While some of the features in this network may originate from cosmetic products, as hypothesized above, the prevalence of microbeMASST hits suggests that many of the features are, in fact, of microbial origin. This supports the hypothesis that fatty acid ethyl esters are products of the microbial metabolism of fatty acids derived from sebum lipids (see above). Such microbial metabolites of host-derived molecules can be seen as products of the holobiont's metabolism.

## Metagenomic sequencing

As a complementary approach for linking metabolites to specific microorganisms, we took advantage of the metagenomic sequencing data set obtained from the same volunteers at the same time as the metabolomics samples (see Materials and Methods for details; data available in the NCBI Sequence Read Archive [SRA] under accession PRJNA1013332). Figure 4A shows the microbiome composition of the individual samples. As expected for the human skin (2), Propionibacteriaceae (mainly *Cutibacterium acnes* plus some reads that were not assigned to the genus level) and different species of the genus *Staphylococcus* were the dominating bacterial taxa. Of note are the large differences in the relative proportions of Propionibacteriaceae and Staphylococci between samples. As expected, Propionibacteriaceae tended to be more abundant in oily scalp samples, but even in this group, some samples were dominated by staphylococci. *S. caprae* or *S. capitis* (species not resolved) generally were the most abundant

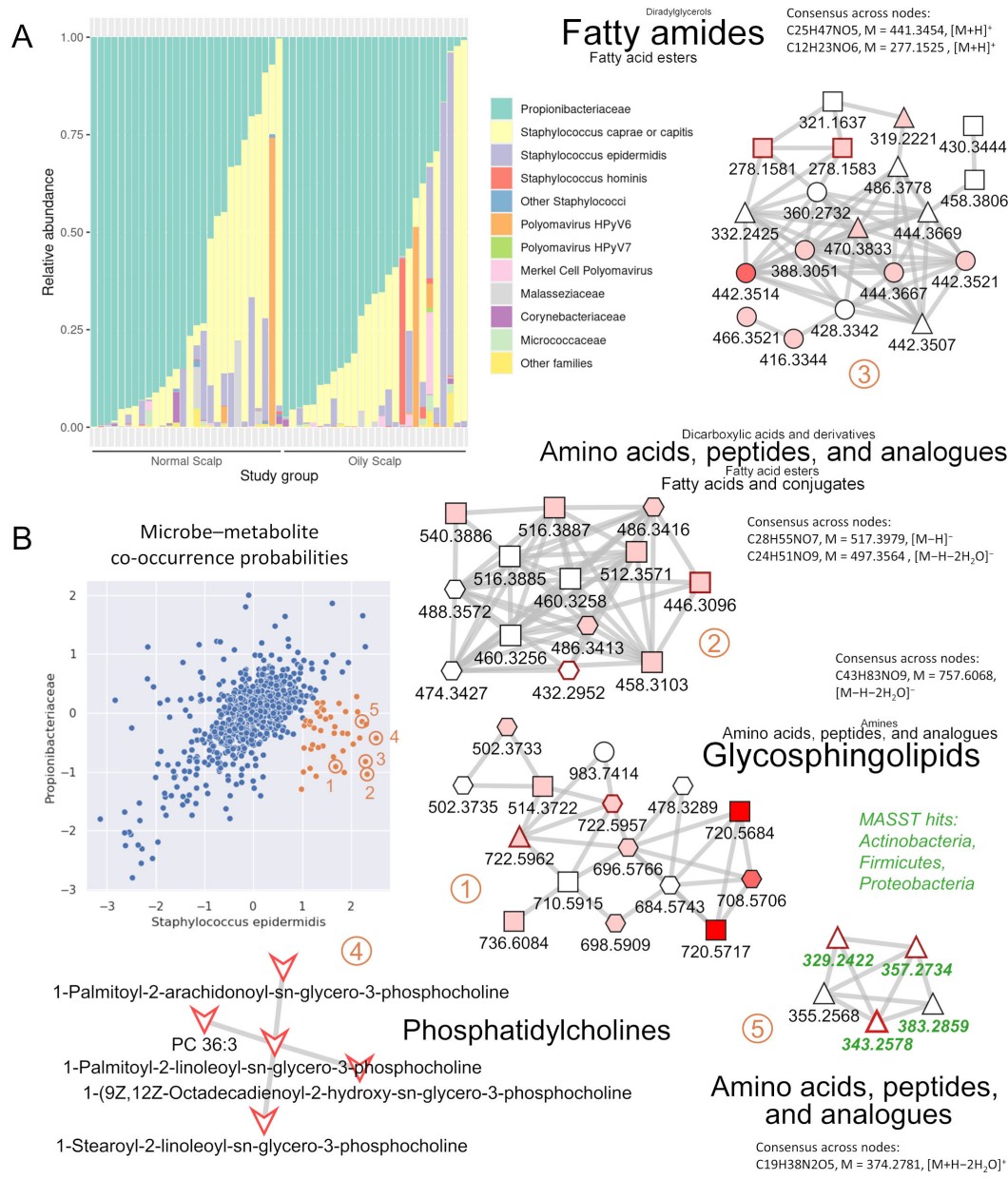

**FIG 4** Metabolomics–microbiome data integration with MMvec. (A) Microbiome data set used as input for MMvec. Normal scalp, Sebumeter score 60–120; oily scalp, Sebumeter score > 120. Samples are arranged in the order of decreasing relative abundance of Propionibacteriaceae. (B) Co-occurrence probabilities (log scale) of molecular families (represented by dots) given the presence of either Propionibacteriaceae or *Staphylococcus epidermidis*. A group of molecular families showing high and specific co-occurrence probabilities with *S. epidermidis* is highlighted in red. A selection of molecular families positively correlated with skin oiliness (families 1–3) or skin moisture (families 4 and 5) is shown in detailed view with GNPS, SIRIUS, and microbeMASST annotations. For a legend of symbols, see Fig. 3.

staphylococci, but some samples contained high proportions of *S. epidermidis*, which is the dominating *Staphylococcus* species in many studies of the human skin from other parts of the body (22). *S. hominis* was the dominating *Staphylococcus* species in only a single sample, while several samples contained *S. hominis* at smaller relative proportions. A few samples showed significant infection with different strains of *Polyomavirus* (HPyV6, HPyV7, Merkel cell polyomavirus). Microbes from the families of Malasseziaceae, Corynebacteriaceae, Micrococcaceae, and further families were found in small relative

proportions in most samples, pointing to a relatively low phylogenetic diversity of the human scalp microbiome in the present study.

## Discovery of microbe–metabolite co-occurrence patterns with MMvec

MMvec predicts the abundance of all metabolites, given the presence of each microbial taxon (6). In doing so, it goes through the microbial taxa one-by-one, each time considering all metabolite features simultaneously. To decrease the risk of overfitting the neural network due to the large number of metabolite features, we performed data reduction on the metabolomics data set, taking advantage of the molecular networks generated by GNPS. We observed that the feature intensities within one molecular network were, in many cases, positively correlated with each other (Fig. S27), presumably due to biological similarities between structurally related metabolites but also due to different features representing the same metabolite (e.g., different adducts). We, therefore, used as input for MMvec the median value of all metabolite features within each molecular network. Considering only networks with at least three connected nodes, the dimensions were reduced to 1,225 molecular networks representing 9,076 out of 44,570 total features.

The output from MMvec is in the form of conditional co-occurrence probabilities, i.e., the probabilities of each molecular network being observed given the presence of a particular microbial taxon. To identify molecular networks specifically co-occurring with one or a subset of the microbial taxa, we plotted pairwise correlations of the conditional probabilities on a log scale for all taxa in the microbiome data set (Fig. 4B; Fig. S28 through S30). Each dot in these figures represents one GNPS molecular network. Most dots aligned in a more or less elongated ellipse along the diagonal of all pairwise comparisons. These were regarded as unspecific co-occurrences. A distinct cluster of molecular networks (red circles) was identified, which had higher co-occurrence probabilities with *S. epidermidis* than with the other microbial taxa (co-occurrence probabilities with *S. epidermidis* and Propionibacteriaceae shown in Fig. 4B). Notably, *S. epidermidis* was not the most abundant *Staphylococcus* species in most samples but was still found in a sufficient number of samples to expect statistically valid results. As a control, the MMvec analysis was repeated three times with the original sample identifiers, giving similar results and three times with scrambled sample identifiers to break the link between the metabolomics and the microbiome data sets, giving negative results (Fig. S31 through S33).

## A metabolic niche for *S. epidermidis*

Figure 4B shows examples of GNPS molecular networks specifically co-occurring with *S. epidermidis*. Molecular networks 1–3 comprise nodes positively correlated with skin oiliness (Sebumeter score), while networks 4 and 5 are positively correlated with skin moisture (Skicon score). Network 4 can be annotated as unsaturated phosphatidylcholines with high confidence; all nodes are direct GNPS library hits. Network 5 comprises short peptides (likely dipeptides). Although this network lacks direct GNPS library hits, the annotation as dipeptides can be inferred with high confidence from the GNPS analog mode hits to di- and tripeptides, the CANOPUS molecular class prediction "Amino acids, peptides, and analogs," and the presence of two nitrogens in the predicted molecular formulas. Four nodes from this network are also microbeMASST hits, indicating a microbial origin of these dipeptides. Most nodes in network 1 are likely glycosphingolipids. The number of oxygen and carbon atoms in the predicted molecular formulas indicates that their chain lengths and degree of hydroxylation match the structure of glucosylceramide species typically found in mammalian cells. Networks 2 and 3 are more challenging to annotate. Network 2 is from the C18− data set and is thus presumably more hydrophobic than network 3, which is from the HILIC+ data set. The molecular formula predictions for network 2 show a single nitrogen and either seven (assuming a $[M-H]^-$ adduct) or nine oxygens (assuming a double water loss $[M-H-2H_2O]^-$). Despite the most common compound class annotation being "Amino

acids, peptides, and analogs," it appears more likely that the nodes from network 2 are lysoglycosphingolipids (e.g., glucopsychosine, $C_{24}H_{47}NO_7$, $[M–H]^- = 460.3274$) and possible longer-chain derivatives. The most common compound class annotation in network 3 is "Fatty amides." Nodes with a predicted molecular formula having a single nitrogen and six oxygens may be *N*-acyl-glucosamines (e.g., N-hexanoyl-beta-D-glucosyl-amine, $C_{12}H_{23}NO_6$, $[M + H]^+ = 278.1604$), which would match the CANOPUS prediction. Nodes with only five oxygens may be hydroxylated acylcarnitines (e.g., 3-hydroxy-octa-decenoyl carnitine, $C_{25}H_{47}NO_5$, $[M + H]^+ = 442.3532$) (23).

The cluster of molecular networks discovered in the MMvec analysis thus defines a metabolic niche for *S. epidermidis* on the human scalp that is enriched for glycosphingo-lipids, phosphatidylcholines, dipeptides, and amide-linked fatty acids. The significance of this finding lies in the observation that *S. epidermidis* is found in major relative proportions in only a subset of the volunteers, with *S. caprae* or *capitis* (species not resolved) generally being the dominant *Staphylococcus* species in the present study. Neither glycosphingolipids nor phosphatidylcholines are typical epidermal surface lipids (24, 25). Glucosylceramides are synthesized by epidermal cells and serve as precursors of stratum corneum ceramides (26). It may thus be speculated that they originate from dermal fibroblasts that could have been exposed at the skin surface, e.g., in wounds (27). The present results may guide the design of follow-up studies specifically aimed at identifying conditions that control the balance between different *Staphylococcus* species on the human skin.

## Final remarks

Our functional metabolomics strategy enabled us to go beyond a mere description of the metabolites present on the human scalp and their relative abundances. Of particu-lar value were microbeMASST and MMvec, tools that allowed us to derive functional metabolite annotations by connecting to microbial data (from publicly available data sets in the case of microbeMASST; from a microbiome data set from the present study in the case of MMvec). microbeMASST results supported the hypothesis that the fatty acid ethyl esters in Fig. 3A are a product of microbial metabolism by showing that these and similar compounds have been observed as microbial metabolites in independent studies under different conditions. MMvec allowed us to discover a metabolic niche for one of the key microbial players on the human skin, *S. epidermidis* (Fig. 4). It is striking that no equally specific co-occurrence pattern could be discerned for *S. caprae/capitis*, although these microbes were more abundant and prevalent than *S. epidermidis* and also showed substantial differences in abundance between samples. The metabolites co-occurring with *S. epidermidis* (Fig. 4) as well as the presumed production of fatty acid ethyl esters from sebum lipids (Fig. 3A) likely represent large-scale activities on the human scalp. More highly powered studies as well as independent experimental confirmation will be required to reveal more subtle details of the roles of metabolites in the interactions between microbes and their human host. Also, given the large variability between microbiome samples (Fig. 4A), sampling from several skin locations per individual would allow a more precise differentiation between inter- and intra-individual variability.

Nevertheless, the present study allowed the generation of hypotheses on physiologi-cal processes that could govern the equilibrium of the skin ecosystem. The functional metabolomics strategy is highly generic and can be conveniently applied to other studies and experimental systems. We anticipate that our methodological approach will be adapted to generate valuable biological insight into the nature of metabolite-microbe interactions from metabolomics and microbiome studies.

## MATERIALS AND METHODS

### Oily scalp study

Study participants from the New York metropolitan area (healthy, aged 19–66 years) were recruited by BASF at the Tarrytown Consumer Testing Center (Tarrytown, NY, USA). Participants were assigned to one of two cohorts during pre-screening: oily scalp cohort (18 male and 58 female participants of different ethnic backgrounds) and normal scalp cohort (32 female participants, mainly Caucasian). Samples for metabolomics and for metagenomic sequencing were taken in August 2018 from the same study participants during the same visit. Skin oiliness (Sebumeter) and moisture (Skicon) were measured during the sampling visit. Samples from both cohorts were combined and re-grouped based on their Sebumeter value: oily scalp group (higher than 120 µg/cm², 12 male and 29 female participants of different ethnic backgrounds) and normal scalp group (between 60 and 120 µg/cm², 2 male and 31 female participants, still mainly Caucasian but more diverse than the original oily scalp cohort). The sample numbers in the metagenomic sequencing data set are lower due to some samples failing the quality check after DNA library preparation (see sample metadata in the Supplementary Information).

Samples for metagenomic sequencing were collected from the participant's scalp using a sterile cotton swab. Two line segments were created (one on each side of the participant's head) by separating the hair fibers with a comb. A sterile cotton swab was moistened with SCF-1 solution (50 mM Tris buffer [pH 7.6], 1 mM EDTA [pH 8.0], and 0.5% Tween-20, NIH Human Microbiome Project—Core Microbiome Sampling Protocol A [HMP-A], https://www.ncbi.nlm.nih.gov/projects/gap/cgi-bin/GetPdf.cgi?id=phd002235.3) and rubbed along the line segment for 60 seconds. One swab was used for each line segment. Both swabs were then inserted into the same tube containing phosphate-buffered saline to release the microbes. The tubes were stored on ice until DNA extraction with the QIAamp DNA Microbiome Kit (QIAGEN). The manufacturer's instructions were followed with the exception that the procedure was interrupted after step 3. Samples were stored at −20°C until the procedure was continued with step 4.

Samples for metabolomics were collected from the participant's scalp using HydraFlock swabs (Puritan) following the previously described protocol (12, 28). A line segment was created in the middle of the participant's head by separating the hair fibers with a comb. A swab was moistened with ethanol/water 1:1 (vol/vol) and rubbed along the line segment for 60 seconds. The tip of the swab was cut and stored in a safe-lock Eppendorf tube at −80°C. The samples were shipped on dry ice to BASF Metabolome Solutions GmbH (Germany), where the metabolomics analysis was performed.

### Metagenomic sequencing

The metagenomic sequencing data set is available in the SRA (PRJNA1013332).

#### Library preparation and sequencing

Whole metagenome sequencing libraries were prepared from 26 µL of DNA solution using the NEBNext Ultra II FS DNA Library Prep Kit (New England Biolabs). The DNA was purified and size selected to remove excess adaptors and adaptor dimers using Ampure XP beads (Beckman Coulter Life Sciences). Adaptor-ligated DNA was then subjected to PCR amplification with universal primers and an index barcode that was unique for each sample. The PCR products were cleaned and size selected with beads to get a 250–600-bp library. The library was checked for quality and quantity on a Bioanalyzer (Agilent) before shipping to a commercial service provider for paired-end sequencing on a HiSeq 3000 (Illumina). The average sequencing depth was 9.9 Gbases per sample.

### Microbiome data processing

After de-multiplexing and quality check, contaminants, adaptors, and known artifacts were removed from the raw reads using bbduk (29). Quality trimming was applied to the ends of reads targeting a Q10 Phred score using bbduk. Reads shorter than 70 bp after trimming were removed. Human reads were removed from the data set by mapping to the GRCh38 assembly using Bowtie 2 (30) with the --sensitive option. All remaining reads were deposited in the SRA under accession no. PRJNA1013332 and processed using MetaPhlAn2 (31) to generate a taxonomic abundance profile for each sample. The resulting table of relative abundances as well as the corresponding sample metadata can be found in the supplemental material.

## Metabolomics analysis

The metabolomics data sets are available on MassIVE (MSV000088283, doi:10.25345/C5DK1G).

### Sample extraction

The swab tips were first extracted with 500 µL of ethanol, followed by two extractions with 500 µL of 2-propanol. Each extraction is comprised of the steps of vortexing, shaking for 5 minutes at 1,400 rpm, 12°C in a ThermoMixer (Eppendorf), and centrifuging for 5 minutes at 12,000 rpm, 12°C. The three extracts from each sample were combined and evaporated in an IR Dancer (Hettlab). The samples were reconstituted with a Bead Ruptor (Omni International) in 1,400 µL of methanol/dichloromethane 2:1 (vol/vol), 120 µL water, 8 µL formic acid, 25 µL polar internal standard solution in water (methionine-D3, tryptophan-D5, arginine-13C6-15N4, Boc-Ala-Gly-Gly-Gly-OH, methyl-alpha-glucopyranoside, ribitol, and alanine-D4, glycine-D2), and 100 µL lipid internal standard solution in methanol/dichloromethane 2:1 (vol/vol) (coenzyme Q1, coenzyme Q2, coenzyme Q4, nonadecanoic acid methyl ester, nonacosanoic acid methyl ester, tridecanoic acid, and pentadecanoic acid). One hundred microliters from each sample was transferred into a glass vial for LC-MS analysis. In addition, an equal amount from each sample was taken, pooled, split into the required number of Pool (technical reference) samples, and included with the LC-MS analysis. Blank samples were generated as further technical controls by extracting tips from unused swabs.

LC-MS/MS data acquisition was performed in a randomized sequence on a TripleTOF 6600 (AB Sciex) by ESI in both positive and negative ionization modes. Each sample was analyzed using both HILIC and C18 chromatography, giving a total of four runs per sample. The HILIC gradient elution was performed with acetonitrile/water/acetic acid 99:1:0.2 (vol/vol/vol, solvent A) and 0.007 M aqueous ammonium acetate/acetic acid 100:0.2 (vol/vol, solvent B) from 0% B to 90% B in 5 minutes at a flow of 600 µL/minute. The C18 gradient elution was performed with water/methanol/0.1 M aqueous ammonium formate/formic acid 49:49:1:0.5 (wt/wt/wt/wt, solvent A) and methyl-tert-butylether/2-propanol/methanol/0.1 M aqueous ammonium formate/formic acid 56:28:14:1:0.5 (wt/wt/wt/wt, solvent B) from 0% B to 25% B in 0.5 minutes and then to 90% B in 5.4 minutes.

### LC-MS/MS data processing

The mass spectrometry data were centroided and converted from the Sciex proprietary format (.wiff) to the open .mzML format using ProteoWizard 3.0 (MSConvert tool [32]). The mzML files were then processed with MZmine toolbox (33) (version 2.37_corr_17.7_kai_merge_2, available on https://github.com/robinschmid/mzmine2/) using the Ion Identity Molecular Networking workflow (34) that allows advanced feature detection for adduct/isotopolog annotations. The MZmine processing was performed on Ubuntu 18.04 LTS 64-bits workstation (Intel Xeon E5-2637, 3.5 GHz, 8 cores, 64 Gb of RAM). The MZmine project, the MZmine batch file (.XML format), and results files (.MGF and .CSV) are available in the MassIVE data set (MSV000088283, doi:10.25345/

C5DK1G). The MZmine batch files contain all the parameters used during the processing for each data sets. In brief, feature detection and deconvolution were performed with the ADAP chromatogram builder (35) and local minimum search algorithm. The isotopologs were regrouped, and the features (specific *m/z* with retention time peaks) were aligned across samples. The aligned feature list was gap filled, i.e., a secondary feature detection step that fills in signals from raw data into gaps of missing features. Filters only retained features with an associated fragmentation spectrum and that occurred in a minimum of 2 files. Retention time peak shape correlation grouped features originating from the same molecule and annotated adduct/isotopolog for SIRIUS. Finally, the feature quantification table results (.CSV) and fragmentation spectral information (.MGF) were exported with the GNPS module for Feature-Based Molecular Networking (FBMN) analysis on GNPS (15) and with SIRIUS export modules (merge MS/MS across samples with weighted average, in sum intensity mode, and 10 ppm maximum tolerance, min cosine 0.7, isolation width 5 Da). Other parameters are available in the MZmine batch file.

## Structural metabolite annotation

The code used for structural metabolite annotation is available at https://github.com/PhilippTernes/Functional-metabolomics-of-the-human-scalp.

### *GNPS library searches and molecular networking*

The results files from MZmine (.MGF and .CSV files) were uploaded to GNPS (https://gnps.ucsd.edu [8]) and analyzed with the feature-based molecular networking workflow (15). The molecular network topology uses the default parameters (max cluster size = 100, top K = 10, and top K filter = on). Spectral library matching was performed against public fragmentation spectra (MS$^2$) libraries on GNPS and the NIST17 library. Molecular networks were visualized using Cytoscape 3.8.1 software (11).

The MS$^2$ spectra were systematically annotated with the SIRIUS computational tools (4) (v. 4.5.3, headless, Linux). Molecular formulas were computed with the SIRIUS module by matching the experimental and predicted isotopic patterns (36) and from fragmentation tree analysis (37) of MS$^2$.

Molecular formula prediction was refined with the ZODIAC module by using Gibbs sampling (16) for fragmentation spectra that were not chimeric spectra or had poor fragmentation.

*In silico* structure annotation using structures from bio databases was done with CSI:FingerID (17). Systematic class annotations were obtained with CANOPUS (9) and used the ClassyFire ontology (38). The parameters for SIRIUS tools were set as follows, for SIRIUS: molecular formula candidates retained (100), molecular formula database (ALL), maximum precursor ion *m/z* computed (750), profile (qtof), *m/z* maximum deviation (25 ppm), and ions annotated with MZmine were prioritized and other ions were considered in the positive ionization mode ([M + H3N + H]+, [M + H]+, [M + K]+, [M + Na]+, [M + H-H2O]+, [M + H-H4O2]+, [M + NH4]+) and in the negative ionization mode ([M-H]−,[M + Cl]−,[M-H2O-H]−,[M-2H2O-H]−,[M + Br]−); for ZODIAC, threshold filter (0.9), minimum local connections (0); for CSI, FingerID: *m/z* maximum deviation (10 ppm) and biological database (BIO); and for CANOPUS, default parameters.

## Functional metabolite annotation

The code used for functional metabolite annotation is available at https://github.com/PhilippTernes/Functional-metabolomics-of-the-human-scalp.

### *Statistical analysis of LC-MS/MS data*

The feature quantification tables from MZmine were filtered to exclude low-quality features based on the quartiles Q1, Q2, and Q3 of the feature intensity distribution across samples. The quartiles were calculated separately for each feature in the Swab, the Blank, and the Pool samples. A feature was excluded for high Blank value if the boxes

(interquartile ranges) of the Blank and the Pool samples were overlapping in a box plot, i.e., $Q3_{Blank} > Q1_{Pool}$. A feature was excluded for high Pool variability if the box of the Pool samples was wider than 0.5 in a box plot on $\log_{10}$ scale [$IQR_{log10} = \log_{10}(Q3_{Pool}) - \log_{10}(Q1_{Pool}) = \log_{10}(Q3_{Pool} / Q1_{Pool}) > 0.5$].

The intensities from the 44,570 remaining features (all four data sets combined) were $\log_{10}$-transformed and normalized to the median of the intensities from the same feature in the Pool samples that were analyzed on the same day. As expected, the resulting intensities showed an approximate normal distribution centered around zero. The intensities were further normalized to the median of the values from all features within each sample and data set to compensate for differences in the amount of sampled material. The resulting values were used for all downstream statistical analyses.

The correlation between the $\log_{10}$-transformed and normalized feature intensities and the physical skin parameters oiliness and moisture was calculated using the *stats::lm* function in R (39) with age, gender, ethnic group, Sebumeter score, and Skicon score as covariates.

### MicrobeMASST

The microbeMASST tool is still in development but is available through a web dashboard (https://masst.gnps2.org/microbemasst; in its testing phase) and as a Python script for batch processing. Matching against samples from >60,000 microbial monocultures was performed with a Python script using the .mgf spectra file from the IIMN workflow. Matches were achieved with 0.05 precursor *m/z* tolerance and fragment *m/z* tolerance, 0.7 cosine threshold, four minimum matched signals, and analog search off. The wider *m/z* tolerances were chosen to include data sets of instruments with lower resolving powers represented within public data. Results were overlaid with the molecular networking using the feature ID provided by MZmine. Briefly, microbeMASST produced tables of spectral matches to specific data files in the public domain, which were then mapped to the curated list of samples with microbe metadata. Those tabular results are then visualized onto trees of the NCBI taxonomy. Interactive HTML/javascript files summarize the results and provide links to other GNPS/MassIVE tools and services, e.g., matching GNPS reference spectra, matching MassIVE data sets, interactive visualization and analysis of MS data files in the GNPS dashboard (40), and spectral mirror plots with the universal spectrum identifier (USI) (41) and the metabolomics spectrum resolver (42). One conclusion from a microbeMASST to a specific NCBI taxonomy is that the target spectrum was previously acquired in another MassIVE data set.

### Microbiome input data for MMvec

Taxonomic abundance profiles generated with MetaPhlAn2 (see above) were aggregated into the taxonomic groups shown in Fig. 4B. Since MMvec accepts real numbers as input but, in its internal statistical procedures, treats them as an approximation of integer counts, the relative abundances were scaled to 12,000 (1,000 times the number of taxa) "pseudo-counts" per sample, converted into the biom format, and used as input for MMvec without further filtering. The original as well as the aggregated table of relative abundances can be found in the supplemental material together with the corresponding sample metadata.

### Metabolomic input data for MMvec

Filtered, $\log_{10}$-transformed, and normalized feature intensities (see above) were subjected to data reduction as follows: features present in molecular networks with less than three nodes were discarded to focus on the larger molecular families in the data sets. For each of the 1,225 molecular networks with at least three nodes, the median value of the feature intensities was calculated within each sample, resulting in a reduced data set with one data point for each molecular network in each sample. The data set was back transformed to a linear scale, scaled to 1,225,000 (1,000 times the number

of molecular networks) "pseudo-counts" per sample for the same reason as above, converted into biom format, and used as input for MMvec without further filtering.

MMvec version 1.06 was installed using pip into a miniconda environment with python = 3.6, tensorflow = 1.15, and scikit-bio = 0.5.2 and further required packages on a laptop computer running Windows Subsystem for Linux with 12 GB of RAM and run with the following parameters: "--min-feature-count 1 --num-testing-examples 15 --summary-interval 1 --learning-rate 0.001 --latent-dim 3 --epochs 102400 --batch-size 4096 --input-prior 1 --output-prior 1." The run time was 15–18 minutes without the GPU being enabled. Model convergence was examined with the TensorBoard interface. A wide range of input and output priors was tested with similar results. Log-scale conditional co-occurrence probabilities were read from the "ranks" output file and visualized as a Seaborn pairplot.

## Annotated metabolite network

All structural and functional metabolite annotations were mapped to the corresponding nodes in the molecular networks (one "graphml" file from GNPS for each metabolomics data set) and visualized using a custom style in Cytoscape (11). The structural annotations from GNPS and SIRIUS were combined into a comprehensive feature annotation metadata file using a custom Python script. The functional annotations from the statistical analysis of the physical skin parameters and from microbeMASST were formatted as additional metadata files using custom R scripts. The metadata files were loaded into Cytoscape and mapped using node identifiers.

## STORMS checklist

The STORMS checklist for this publication can be found at https://github.com/PhilippTernes/Functional-metabolomics-of-the-human-scalp/blob/6d522dee076777e85b9882f9eba60096528fb0f2/STORMS%20Checklist.xlsx.

## ACKNOWLEDGMENTS

The authors thank Ina Erbe, Nicole Rocker, and Peter Driemert at BASF Metabolome Solutions GmbH for the generation of the mass spectrometry data, James T. Morton at the University of California, San Diego, for the help with the MMvec analysis, and the BASF California Research Alliance (CARA) for the funding.

L.-F.N., P.C.D., R.F., and P.T. conceptualized the multi-omics strategy. A.G., S.L.-O., and V.A.-F. conceptualized the scalp study, L.-F.N. performed the mass spectrometry data processing and structural metabolite annotation. P.T. performed the statistical analysis of physical skin parameters. R.S. performed the microbeMASST analysis. H.C. performed metagenomics sequencing and data processing. L.-F.N. and P.T. performed the MMvec analysis. L.-F.N. and P.T. visualized and interpreted the results and wrote the manuscript. P.C.D. and P.T. supervised the work presented in the paper.

## AUTHOR AFFILIATIONS

[1]Collaborative Mass Spectrometry Innovation Center, University of California San Diego, La Jolla, California, USA

[2]Skaggs School of Pharmacy and Pharmaceutical Sciences, University of California San Diego, La Jolla, California, USA

[3]Institute of Organic Chemistry and Biochemistry of the Czech Academy of Sciences, Prague, Czechia

[4]BASF Corporation, Tarrytown, New York, USA

[5]BASF Corporation, Research Triangle Park, North Carolina, USA

[6]BASF Beauty Care Solutions France S.A.S, Lyon Cedex, France

[7]BASF Metabolome Solutions GmbH, Berlin, Germany

## AUTHOR ORCIDs

Philipp Ternes  http://orcid.org/0000-0003-0886-5474

## FUNDING

| Funder | Grant(s) | Author(s) |
|---|---|---|
| BASF | BASF Corporation | California Research Alliance (CARA) | Louis-Félix Nothias |

## DATA AVAILABILITY

The metagenomic sequencing data set is available in the SRA (PRJNA1013332). The metabolomics data sets are available on MassIVE (MSV000088283, doi:10.25345/C5DK1G). Reads were deposited in the SRA under accession no. PRJNA1013332.

## ETHICS APPROVAL

The protocol for the present study (# TC-0118-003-095 A and B) is based on Parent Protocol "Hair/Scalp Study" (# TC-0315-001-006), which was reviewed and approved by the BASF Tarrytown Consumer Testing Center Institutional Review Board (TCTC IRB). The protocol also covers the collection, sequencing, and data analysis of microbiome samples. Informed consent was obtained from all participating volunteers, and the experiments conformed to the principles set out in the WMA Declaration of Helsinki and the Department of Health and Human Services Belmont Report.

## ADDITIONAL FILES

The following material is available online.

### Supplemental Material

**Supplemental Information (mSystems00356-23-S0001.pdf).** Table S1 and supplemental figures.
**Table S2 (mSystems00356-23-S0002.xlsx).** Sample metadata for the metabolomics and metagenomic sequencing data sets.
**Table S3 (mSystems00356-23-S0003.xlsx).** Relative taxon abundances derived from the metagenomic sequencing data set (MetaPhlAn2).
**Table S4 (mSystems00356-23-S0004.xlsx).** STORMS microbiome reporting checklist.

### Open Peer Review

**PEER REVIEW HISTORY (review-history.pdf).** An accounting of the reviewer comments and feedback.

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
