## [Reviewer comments · mSystems]

Functional metabolomics of the human scalp: A metabolic niche for *Staphylococcus epidermidis*

Louis-Félix Nothias, Robin Schmid, Allison Garlet, Hunter Cameron, Sabrina Leoty-Okombi, Valérie André-Frei, Regine Fuchs, Pieter Dorrestein, and Philipp Ternes

Corresponding Author(s): Philipp Ternes, BASF Metabolome Solutions GmbH

Review Timeline:

Submission Date:	April 24, 2023
Editorial Decision:	May 31, 2023
Revision Received:	October 4, 2023
Accepted:	November 29, 2023

Editor: Morgan Langille

Reviewer(s): The reviewers have opted to remain anonymous.

Transaction Report:

DOI: <https://doi.org/10.1128/msystems.00356-23>

May 31, 2023

Dr. Philipp Ternes
BASF Metabolome Solutions GmbH
Berlin
Germany

Re: mSystems00356-23 (A multi-omics strategy for the study of microbial metabolism: application to the human skin's microbiome)

Dear Dr. Philipp Ternes:

Thank you for submitting your manuscript to mSystems. We have completed our review and I am pleased to inform you that, in principle, we expect to accept it for publication in mSystems. However, acceptance will not be final until you have adequately addressed the reviewer comments.

Preparing Revision Guidelines

Please return the manuscript within 60 days; if you cannot complete the modification within this time period, please contact me. If you do not wish to modify the manuscript and prefer to submit it to another journal, please notify me of your decision immediately so that the manuscript may be formally withdrawn from consideration by mSystems.

Sincerely,

Promi Das

Editor, mSystems

Journals Department
Reviewer comments:

Reviewer #1 (Comments for the Author):

1. Comment: it is recommended that the authors submit a STORMS Microbiome Reporting Checklist as this work was done on human samples and provides additional clarity on the analytical results being reported.
2. Comment: In the provided supplementary file, it is extremely challenging to interpret the taxonomy identifiers. It is suggested authors describe in more detail how the supplementary data file supports the metabolomics analysis submitted to MassIVE and how the metadata aligns as mentioned in the legend key. It is unclear the value this xlsx file is providing as there are no direct linkages between the omics data files that are apparent. The identifier supplied do not align to the identifiers used in the supplementary image files.
3. Suggestion: It is suggested that the authors include the DOI (<https://doi.org/doi:10.25345/C5DK1G>) provided at MassIVE in the results/discussion section, as it provides a method of reuse/citation.
4. Comment: SI figures appear distorted in uploaded version and require revised upload. Additionally, the microbial identifiers and the output molecules are not clearly provided in the supplementary xlsx file (of which the microbial taxonomy identifiers outlined in the manuscript and SI images did not appear to match).

Reviewer #2 (Comments for the Author):

Nothias et al use multiple 'omics' approaches to investigate the relationship between metabolites and microorganisms found on human subjects with oily vs. 'normal' scalps. The strategy used by the authors enabled the identification of potential antagonistic interactions between two members of the scalp microbiota, *S. epidermidis* and *C. acnes*. The manuscript has good potential to guide how meta-'omic' datasets are analyzed by the microbiology/microbial ecology community but there are substantial fundamental issues with the manuscript that should be addressed/clarified to make the manuscript better and accessible to readers in the fields of microbiology & microbial ecology.

1. The first issue with the manuscript has to do with the goal/purpose of the manuscript. The manuscript appears to be largely written as a metabolomics methods paper with some microbial element included at the end as a proof of concept. In its current form the manuscript is beyond the scope and comprehension of most mSystems readers. The bulk of the manuscript focuses on the differences between the different analytical methods used and not the microbial findings of the study. The authors need to decide if this is a metabolomics/metagenomics methods paper or a research paper that summarizes the findings of their oily vs. normal scalp study and adjust the writing accordingly.
2. The manuscript emphasizes on nodes, features, molecular features, networks etc. but the biological interpretations are not well described. The authors could focus on the main biological findings of their manuscript to make the manuscript accessible to the average microbiology/ microbial ecology reader.
3. It is also not entirely clear to me how the microbiome and metabolomics datasets were integrated. This is the main highlight of the manuscript and should be explained in a way that can be understood by readers of the journal. The authors have not demonstrated exactly how relationships between metabolites and microorganisms were established. The production of metabolites by microbes is influenced by several factors many of which are not accounted for by the authors methods/algorithms. Just because a microbe can produce a metabolite under certain conditions does not mean it will always produce that metabolite. The current description of microbe-metabolite assignments (through co-associations) sounds like it will lead to a lot of false positive microbe-metabolite assignments. Other relevant questions are: i) How do the authors link metabolites produced by multiple microbes to an individual microbe when multiple producers are present? ii) How is metabolite production scaled between multiple producers?
4. There are fundamental sampling issues which preclude the authors from making any robust interpretations since technically two different areas of the scalp were sampled for metabolomic and metagenomic analysis. The authors swabbed two different areas of the scalp which had potentially different microbes and metabolite profiles and are linking metabolite production from one area of the scalp with metabolite profiles at another area of the scalp. Repeat sampling from the same location on the scalp for different tests (metabolomics & metagenomics) will result in a disruption of the microbial community/metabolites accumulated in

that area of the scalp, which further brings to question the robustness of the data collected.

5. It is not clear to me how positive/negative correlations between molecular families and specific microbes led to the conclusion that antagonistic interactions exist between *S. epidermidis* and *C. acnes*. The authors should elaborate on this since it is a major finding of the study.

6. A few areas of the manuscript use vague terminology and phrases that are challenging to comprehend .e.g. "The action is missing", "finally revealing the action in our story" etc. Could the authors address this?

Reviewer #3 (Comments for the Author):

"A multi-omics strategy for the study of microbial metabolism: application to the human skin's microbiome" by Nothias and colleagues describes a multiomics integration strategy bringing together mass spec data and microbial abundances. They leverage several existing tools and resources including the Global Natural Products Social (Molecular Networking), a modified MASST search (microbialMASST) and the MMvec tool. They apply this framework to a sebaceous body site, presumably chosen because sebaceous sites tend to be lower diversity, to generate a integrated network view of microbes and metabolites.

I don't like to start with an "editorial" comment but this isn't really a multiomics study. Well over half the paper's text and every single figure, including the supplemental figures, is mass-spec data. There's not a single barplot, PCoA or main table describing the metagenomics (there is a supplemental table of relative abundances). Its not a generalized "multi-omics strategy" as the title suggests. This doesn't have to be a criticism though, I think the authors should lean into it. It's a great example of how mass-spec provides a complementary view of the microbiome. It's also a good example of how to rigorously do such an experiment (e.g., combining multiple conditions, etc...). I learned a lot as someone who has watched mass-spec for the microbiome from the sidelines. I suggest the authors "rebrand" this as a mass-spec paper, because the mass-spec is well done and interesting.

Figure 1 suggests a metagenome assembled genomes approach (yellow box) but the methods say metaphlan2. This is a huge methodological difference and, again, points to the metagenomics being an afterthought rather than an integral part of the analysis. Please update Figure 1.

The scalp lies on the less complex end of the skin microbiome spectrum, it's highly enriched in Cutibacteria and Malassezia. Even with that constraint, none of the 3 molecular networks between *C.acn* and *S. epi* could be fully resolved and the source of the metabolites is speculative (pg 8). The limited information about metabolite source is from the microbialMASST analysis, which only uses mass-spec data (pg 7, top). The remaining network connections are largely/entirely ignored, including an interesting looking Polyomavirus HPyV6 connected region. The paper also ignores the *Malassezia globosa* portion of the network (lower right, fig 4a) which is likely the most clinically relevant part of the scalp microbiome. Why are there two *s_Cutibacterium_acnes* nodes in the Fig 4 graph; it appears to derive from two different *C. acnes* strains resolved by Metaphlan2 but isn't discussed? *S. capitis* is the second most abundant (and most prevalent, in 55/56 samples) microbe in the relative abundance table but it isn't discussed. The authors need to show how this an integrated analysis that is more than the sum of it's parts (published databases like GPNS and tools like MMvec) with a detailed analysis of the final interactome.

This this is meant to be a roadmap for others, a github repo documenting the integrated analysis should be provided.

Where is the Data Availability section? The metagenomic data needs to be deposited in the SRA and, if MAGs were generated it would be nice if those were made available.

Minor comments:

- * GPNS, define first use
- * SCF-1 solution, define first use
- * What depth were metagenomes sequenced to on the HiSeq3000?
- * Sample EMR_04_N-22_RD is 100% unclassified in the supplemental relative abundance table

Reviewer #1 (Comments for the Author):

1. Comment: it is recommended that the authors submit a STORMS Microbiome Reporting Checklist as this work was done on human samples and provides additional clarity on the analytical results being reported. – *The STORMS Checklist is included as Supplementary Information as well as online on GitHub.*

2. Comment: In the provided supplementary file, it is extremely challenging to interpret the taxonomy identifiers. It is suggested authors describe in more detail how the supplementary data file supports the metabolomics analysis submitted to MassIVE and how the metadata aligns as mentioned in the legend key. It is unclear the value this xlsx file is providing as there are no direct linkages between the omics data files that are apparent. The identifier supplied do not align to the identifiers used in the supplementary image files. – *All required information is present. The legends are revised to better explain the links between files.*

3. Suggestion: It is suggested that the authors include the DOI (<https://doi.org/doi:10.25345/C5DK1G>) provided at MassIVE in the results/discussion section, as it provides a method of reuse/citation. – *The DOI is now included in the results/discussion section.*

4. Comment: SI figures appear distorted in uploaded version and require revised upload. Additionally, the microbial identifiers and the output molecules are not clearly provided in the supplementary xlsx file (of which the microbial taxonomy identifiers outlined in the manuscript and SI images did not appear to match). – *There probably is a misunderstanding here. The microbial taxonomy identifiers in the SI images are from microbeMASST and thus point to publically available datasets in MASSIVE, while the taxonomy identifiers in the supplementary xlsx file are from the MetaPhlan analysis. There is thus no direct connection between the two. Ultimately, both taxonomies are derived from the NCBI taxonomy.*

Reviewer #2 (Comments for the Author):

Nothias et al use multiple 'omics' approaches to investigate the relationship between metabolites and microorganisms found on human subjects with oily vs. 'normal' scalps. The strategy used by the authors enabled the identification of potential antagonistic interactions between two members of the scalp microbiota, *S. epidermidis* and *C. acnes*. The manuscript has good potential to guide how meta-'omic' datasets are analyzed by the microbiology/microbial ecology community but there are substantial fundamental issues with the manuscript that should be addressed/clarified to make the manuscript better and accessible to readers in the fields of microbiology & microbial ecology.

1. The first issue with the manuscript has to do with the goal/purpose of the manuscript. The manuscript appears to be largely written as a metabolomics methods paper with some microbial element included at the end as a proof of concept. In its current form the manuscript is beyond

the scope and comprehension of most mSystems readers. The bulk of the manuscript focuses on the differences between the different analytical methods used and not the microbial findings of the study. The authors need to decide if this a metabolomics/metagenomics methods paper or a research paper that summarizes the findings of their oily vs. normal scalp study and adjust the writing accordingly. – *Large parts of the manuscript are now rewritten to provide a clearer focus. The intended focus is a conceptual methods paper that shows how untargeted metabolomics, including state-of-the-art data analysis tools, can be applied in a microbiology context. While primarily metabolomics data is used to address the microbiology research question, metagenomics data comes in as ‘functional annotations’ to enhance the metabolomics data.*

2. The manuscript emphasizes on nodes, features, molecular features, networks etc. but the biological interpretations are not well described. The authors could focus on the main biological findings of their manuscript to make the manuscript accessible to the average microbiology/microbial ecology reader. – *The intended focus is a conceptual methods paper on applying untargeted metabolomics, so the use of some technical terms cannot be avoided. Of course, we hope that the presented biology (‘A metabolic niche for Staphylococcus epidermidis’) will also be interesting to more biology-focused readers and illustrate the value that untargeted metabolomics can provide. We are confident that the accessibility of the text has also improved during revision.*

3. It is also not entirely clear to me how the microbiome and metabolomics datasets were integrated. This is the main highlight of the manuscript and should be explained in a way that can be understood by readers of the journal. The authors have not demonstrated exactly how relationships between metabolites and microorganisms were established. The production of metabolites by microbes is influenced by several factors many of which are not accounted for by the authors methods/algorithms. Just because a microbe can produce a metabolite under certain conditions does not mean it will always produce that metabolite. The current description of microbe-metabolite assignments (through co-associations) sounds like it will lead to a lot of false positive microbe-metabolite assignments. Other relevant questions are: i) How do the authors link metabolites produced by multiple microbes to an individual microbe when multiple producers are present? ii) How is metabolite production scaled between multiple producers? – *In the revised manuscript, we introduce the microbiome data more cautiously as an enhancement (‘functional annotations’) of the metabolite data. The algorithms used, microbeMASST and MMvec, are of course still the same. “Just because a microbe can produce a metabolite under certain conditions does not mean it will always produce that metabolite.”: This is indeed a limitation of the microbeMASST approach. If a metabolite is a microbeMASST hit, it just indicates that it might be of microbial origin. There is no proof that the respective metabolite is of microbial origin in the present study. We hope that presenting microbeMASST hits as ‘functional annotations’ is sufficiently cautious. “How do the authors link metabolites produced by multiple microbes to an individual microbe when multiple producers are present?”: This is achieved through the algorithm in MMvec, which statistically considers metabolite and microbe abundances in individual samples. Of course, the limited number of samples in the present studies allows only for discovering big trends. The discovery of more subtle associations would require more highly powered studies. “How is metabolite production scaled between multiple*

producers”: If we understand this question correctly as asking for the relative contributions of individual producers to metabolites that are being produced by several microbes, this question is far more complex than MMvec was designed for, but in principle could be answered by a similar type of algorithm (also given sufficient sample numbers and sufficient meaningful biological variability in the dataset). It is beyond the scope of the present study. We hope the present study presents an initial step in the right direction.

4. There are fundamental sampling issues which preclude the authors from making any robust interpretations since technically two different areas of the scalp were sampled for metabolomic and metagenomic analysis. The authors swabbed two different areas of the scalp which had potentially different microbes and metabolite profiles and are linking metabolite production from one area of the scalp with metabolite profiles at another area of the scalp. Repeat sampling from the same location on the scalp for different tests (metabolomics & metagenomics) will result in a disruption of the microbial community/metabolites accumulated in that area of the scalp, which further brings to question the robustness of the data collected. – *The reviewer is right that this is a limitation of how sampling was performed in the present study. As the reviewer notes, the alternative of repeated sampling from the same location would have been no better, so we performed the study the way we did. The only way out would be sampling from multiple locations and analyzing these samples individually (in the present study, the microbiome was sampled from two locations per individual but the samples were pooled). This would also not solve the problem but would at least allow an assessment of the variability between locations. We have included a corresponding recommendation in the discussion section.*

5. It is not clear to me how positive/negative correlations between molecular families and specific microbes led to the conclusion that antagonistic interactions exist between *S. epidermidis* and *C. acnes*. The authors should elaborate on this since it is a major finding of the study. – *This part of the manuscript is now rewritten and the data are presented differently. The reviewer is right that this was a limitation in the previous version, which we hope is now resolved.*

6. A few areas of the manuscript use vague terminology and phrases that are challenging to comprehend .e.g. "The action is missing", "finally revealing the action in our story" etc. Could the authors address this? – *We have taken care to use clearer language in the revised version.*

Reviewer #3 (Comments for the Author):

"A multi-omics strategy for the study of microbial metabolism: application to the human skin's microbiome" by Nothias and colleagues describes a multiomics integration strategy bringing together mass spec data and microbial abundances. They leverage several existing tools and resources including the Global Natural Products Social (Molecular Networking), a modified MASST search (microbialMASST) and the MMvec tool. They apply this framework to a sebaceous body site, presumably chosen because sebaceous sites tend to be lower diversity, to generate a integrated network view of microbes and metabolites.

I don't like to start with an "editorial" comment but this isn't really a multiomics study. Well over half the paper's text and every single figure, including the supplemental figures, is mass-spec data. There's not a single barplot, PCoA or main table describing the metagenomics (there is a supplemental table of relative abundances). It's not a generalized "multi-omics strategy" as the title suggests. This doesn't have to be a criticism though, I think the authors should lean into it. It's a great example of how mass-spec provides a complementary view of the microbiome. It's also a good example of how to rigorously do such an experiment (e.g., combining multiple conditions, etc...). I learned a lot as someone who has watched mass-spec for the microbiome from the sidelines. I suggest the authors "rebrand" this as a mass-spec paper, because the mass-spec is well done and interesting. – *We have rebranded the manuscript as a conceptual methods paper on applying untargeted metabolomics and revised large parts of the text accordingly. Microbiome data is still being brought in through the microbeMASST and MMvec algorithms, but now serves as 'functional annotations' to the metabolome data. We hope that this shift in focus better matches the presented data.*

Figure 1 suggests a metagenome assembled genomes approach (yellow box) but the methods say metaphlan2. This is a huge methodological difference and, again, points to the metagenomics being an afterthought rather than an integral part of the analysis. Please update Figure 1. – *Figure 1 has been updated to match the 'functional metabolomics strategy' presented at the beginning of the Results and Discussion section. The reviewer's comment has been addressed.*

The scalp lies on the less complex end of the skin microbiome spectrum, it's highly enriched in Cutibacteria and Malassezia. Even with that constraint, none of the 3 molecular networks between C.acn and S. epi could be fully resolved and the source of the metabolites is speculative (pg 8). The limited information about metabolite source is from the microbialMASST analysis, which only uses mass-spec data (pg 7, top). The remaining network connections are largely/entirely ignored, including an interesting looking Polyomavirus HPyV6 connected region. The paper also ignores the Malassezia globosa portion of the network (lower right, fig 4a) which is likely the most clinically relevant part of the scalp microbiome. Why are there two s_Cutibacterium_acnes nodes in the Fig 4 graph; it appears to derive from two different C. acnes strains resolved by Metaphlan2 but isn't discussed? S. capitis is the second most abundant (and most prevalent, in 55/56 samples) microbe in the relative abundance table but it isn't discussed. The authors need to show how this is an integrated analysis that is more than the sum of its parts (published databases like GPNS and tools like MMvec) with a detailed analysis of the final interactome. – *"none of the 3 molecular networks between C. acn and S. epi could be fully resolved and the source of the metabolites is speculative": We have redesigned and re-run the MMvec analysis. The revised Figure 4 provides examples of molecular networks which are much better resolved. "The remaining network connections are largely/entirely ignored, including an interesting looking Polyomavirus HPyV6 connected region.": We think that many of the interactions in the previous version of Figure 4 (including Polyomavirus HPyV6) had weak statistical support, so we focused on the three highlighted molecular networks. The revised approach to the MMvec analysis focuses on one trend that we are confident has sufficient*

statistical support and highlights only those interactions from the beginning (revised Figure 4B). The three molecular networks that were highlighted in the manuscript's previous version are among the red circles in the revised Figure 4B, which is reassuring. The revised version highlights different molecular networks, though, as better annotated examples are now available. "The paper also ignores the *Malassezia globosa* portion of the network (lower right, fig 4a), which is likely the most clinically relevant part of the scalp microbiome.": Figure 4A shows that a significant proportion of *Malassezia globosa* reads were detected in only 4–5 individuals, which is too few to draw statistically solid conclusions. This is why we are not highlighting it in the revised version, either. "Why are there two *s_Cutibacterium_acnes* nodes in the Fig 4 graph; it appears to derive from two different *C. acnes* strains resolved by MetaPhlan2 but isn't discussed?" Reads were resolved by MetaPhlan2 to different taxonomic depths. Some reads were resolved down to strain level while others were only resolved to the Family level. During revision, we realized that providing strain-level resolution in the MMvec input data does not make sense if many reads from the same family are not assigned down to strain level. We therefore curated the input data as shown in Figure 4A to a depth that could be reasonably well supported for the respective taxa. "*S. capitis* is the second most abundant (and most prevalent, in 55/56 samples) microbe in the relative abundance table but it isn't discussed.": This is true, but one can see from Supplemental Figures S28–S30 that no specific pattern is visible for *S. capitis* that would be different from the pattern observed to other microbes. This is why we are focusing on *S. epidermidis*. This is also surprising, and a corresponding note has been added to the Results and Discussion part under Final Remarks. "The authors need to show how this an integrated analysis that is more than the sum of its parts (published databases like GNPS and tools like MMvec) with a detailed analysis of the final interactome.": The revised and rebranded manuscript comes with a little more understatement concerning the integrated analysis, and we are confident that it lives up to the expectations it raises.

This this is meant to be a roadmap for others, a github repo documenting the integrated analysis should be provided. – *A GitHub repository has been set up at <https://github.com/PhilippTernes/Functional-metabolomics-of-the-human-scalp>*

Where is the Data Availability section? The metagenomic data needs to be deposited in the SRA and, if MAGs were generated it would be nice if those were made available. – *The metagenomic data is now available at <https://www.ncbi.nlm.nih.gov/bioproject/PRJNA1013332>. No MAGs were generated.*

Minor comments: – *These issues were addressed during revision.*

* GPNS, define first use – *done*

* SCF-1 solution, define first use – *done*

* What depth were metagenomes sequenced to on the HiSeq3000? – *9.9 Gbases per sample on average, now indicated in the Methods sections*

* Sample EMR_04_N-22_RD is 100% unclassified in the supplemental relative abundance table – *this sample was removed from the relative abundance table. It was an obvious sequencing failure.*

Re: mSystems00356-23R1 (Functional metabolomics of the human scalp: A metabolic niche for *Staphylococcus epidermidis*)

Dear Dr. Philipp Ternes:

Your manuscript has been accepted, and I am forwarding it to the ASM production staff for publication. Your paper will first be checked to make sure all elements meet the technical requirements. ASM staff will contact you if anything needs to be revised before copyediting and production can begin. Otherwise, you will be notified when your proofs are ready to be viewed.

Featured Image Submissions: If you would like to submit a potential Featured Image, please email a file and a short legend to mSystems@asmusa.org. Please note that we can only consider images that (i) the authors created or own and (ii) have not been previously published. By submitting, you agree that the image can be used under the same terms as the published article. File requirements: square dimensions (4" x 4"), 300 dpi resolution, RGB colorspace, TIF file format.

Sincerely,
Morgan Langille
Editor
mSystems

Reviewer #1 (Comments for the Author):

Author revisions to the main text have greatly improved the scope and readability of the submitted article. Requested adjustments have been addressed and is now suitable for publication.

Reviewer #2 (Comments for the Author):

The authors have addressed all my concerns. The manuscript reads much better now.

Reviewer #3 (Comments for the Author):

The authors have addressed my original concerns with changes to the text and additional analyses.

Minor comment:

* I didn't see an explanation for why metagenomic data were generated for only 55 of the 74 subjects. It is possible that I just missed it.

* Thank you for depositing the data at SRA and creating a GitHub repo of the analysis